# Socioeconomic inequalities, substance use, and chronic multimorbidity in Kiambu and Nakuru counties in Kenya

Linnet Ongeri[ID][1]*, Lydia Kaduka[2], Damaris Matoke[3], Doris Njomo[4], Zipporah Bukania[2], Moses Mwangi[2], Linus Ndegwa[1], Joanna Olale[ID][1], Caleb Othieno[5], Sahara Hussein[1], Geoffrey Barini[ID][6], Valentine Wanga[7], Polycarp Mogeni[ID][1]

1 Centre for Clinical Research, Kenya Medical Research Institute, Nairobi, Kenya, 2 Centre for Public Health Research, Kenya Medical Research Institute, Nairobi, Kenya, 3 Centre for Biotechnology Research and Development, Kenya Medical Research Institute, Nairobi, Kenya, 4 Eastern and Southern Africa Centre of International Parasite Control, Nairobi, Kenya, 5 Department of Medicine, Maseno University, Kisumu, Kenya, 6 Department of Pure and Applied Mathematics, Jomo Kenyatta University of Agriculture and Technology, Nairobi, Kenya, 7 School of Public Health, University of Washington, Seattle, Washington, United States of America

* longeri@kemri.go.ke

## Abstract

Multimorbidity, defined as the coexistence of two or more chronic conditions in an individual, is a growing public health concern associated with increased healthcare costs, poorer quality of life, and increased mortality. Substance use, defined as the consumption of alcohol, tobacco, or other psychoactive substances, may influence the development of multimorbidity, but evidence from sub-Saharan Africa remains limited. We examined the prevalence of multimorbidity and its association with sub-stance use, as well as socioeconomic and demographic factors, in two counties in central Kenya. Data were obtained from a household-based, cross-sectional survey using a stratified, two-stage random sampling design. To account for the complex survey design, we employed sample-weighted multivariable analysis using a modified Poisson regression model to estimate adjusted prevalence ratios (aPR) and 95% confidence intervals (CIs) for the associations. Among the 1,484 participants (median age, 36 years), 58% were female, 61% were married, and 52% reported lifetime substance use. Multimorbidity was prevalent in 7.7% of study participants and rela-tively higher among participants aged 36–45 years (aPR, 4.32 [95% CI, 1.15–16.28], P = 0.031) and 46 years or older (aPR, 12.52; 95% CI, 3.68–42.56; P < 0.001), female (aPR, 1.62 [95%CI, 1.16–2.27]; P = 0.005), those that were divorced or separated (aPR, 2.22 [95%CI, 1.37–3.62]; P = 0.002) and alcohol users (aPR, 1.68 [95%CI, 1.07–2.64]; P = 0.024). Multimorbidity prevalence did not vary significantly by levels of education or income. These findings underscore the need for integrated, community-based approaches to prevent and manage multimorbidity, particularly among older adults, women, and individuals with a history of alcohol use.

**Data availability statement:** Source data are provided in the supplementary materials.

**Funding:** This study was funded by the Kenya Medical research Internal Research Grants (grant number L 0178 to L.O). The funders had no role in study design, data collection and analysis, decision to publish, or preparation of the manuscript.

**Competing interests:** The authors have declared that no competing interests exist.

## Introduction

Multimorbidity—the coexistence of two or more chronic health conditions in an individual—is an emerging global health priority [1]. Its prevalence has markedly risen over the past two decades, partly driven by aging populations, lifestyle changes, and improved survival resulting from advances in clinical practices worldwide [2]. The *Global Burden of Disease (GBD) 2019* study highlights that multimorbidity accounts for a growing share of Disability-Adjusted Life Years (DALYs) worldwide, with non-communicable diseases (NCDs) responsible for the majority of years lived with disability [3]. This shift has significant implications for health systems that are traditionally structured around managing individual diseases. Whilst single conditions can be managed with standardized, vertical programs, multimorbidity demands integrated, patient-centered approaches that account for interactions among multiple conditions, social determinants of health, and behavioral risk factors [1].

In high-income countries, multimorbidity has long been associated with increased healthcare costs, fragmented care, polypharmacy, and poorer quality of life [4]. However, in low- and middle-income countries (LMICs), especially in sub-Saharan Africa (SSA), the challenges are compounded by fragile health systems, dual burdens of infectious and chronic diseases, and scarce population-level data [5]. Health programs in SSA remain largely disease-specific, reflecting donor priorities and historic public health strategies. This approach leaves little room for managing overlapping conditions that often share common risk factors such as poverty, poor nutrition, and substance use [6]. At the individual level, patients face inconsistent care, high out-of-pocket costs, limited access to long-term management, and a high mortality rate. At the policy level, the absence of robust surveillance data limits governments from designing evidence-informed interventions to address multimorbidity holistically [7,8].

Although Kenya's Social Health Insurance Fund (SHIF) acknowledges the burden of chronic illnesses through initiatives such as the emergency and chronic disease fund, it continues to face implementation challenges,[9] and there are no clear national guidelines or frameworks addressing multimorbidity as a distinct health challenge. NCDs account for over 30% of deaths nationally and contribute significantly to rising DALYs [10]. Yet most research and policy guidelines address these diseases individually, with minimal focus on their co-occurrence or interaction with behavioral risk factors [11]. Substance use, particularly harmful alcohol consumption and tobacco use, is well established as a driver of chronic illnesses such as cardiovascular disease, liver disease, and cancer [12]. Small-scale studies in Nairobi and Kisumu link alcohol use to hypertension and liver disease, while national surveys associate tobacco use with cardiovascular risk [13,14]. Besides behavioral risks, individuals with lower socioeconomic status experience an earlier onset and more rapid accumulation of chronic conditions. For example, a longitudinal study from the United Kingdom found that socioeconomically disadvantaged individuals developed multimorbidity earlier in life and at a higher rate than individuals with higher socioeconomic status [15]. In addition, a recent meta-analysis demonstrated that people with lower educational attainment were associated with higher odds of multimorbidity

compared with those with higher education levels [16]. Yet, despite these consistent findings from high- and middle-income countries, data from low-income settings in SSA remain scarce, leaving major gaps in understanding the association of substance use and social disadvantage with multimorbidity.

This population-based cross-sectional study conducted in two populous Kenyan counties investigates the relationship between multimorbidity—the co-occurrence of depression, hypertension, diabetes, obesity, or underweight—and its association with substance use and socioeconomic status. The findings will provide crucial evidence for policy development aimed at mitigating the growing public health burden of multimorbidity.

## Materials and methods

### Ethical statement

Ethical approval was obtained from the KEMRI Scientific and Ethics Review Unit (KEMRI; SERU #3254). The study was conducted in accordance with the World Medical Association Declaration of Helsinki. All participants were comprehensively informed about the study's purpose, risks, benefits, and procedures. Written informed consent was then obtained from all participants prior to data collection.

### Study setting

The survey included adults from Kiambu and Nakuru Counties, Kenya (Fig 1). Kiambu County, located in central Kenya and bordering Nairobi to the north, has a population of approximately 2.4 million residents and covers 2,543 km²; 60% of the residents reside in urban areas [17]. Nakuru County is 90 km northwest of Nairobi in the Rift Valley, has 2.16 million residents, making it the third most populous county after Nairobi and Kiambu [17]. It covers 7,495 km² and comprises 11 constituencies and 13 administrative towns (Fig 1). These counties were selected for their urban–rural balance and ethnic and linguistic diversity; Kiswahili and Kikuyu are the predominant languages.

### Study participants

Adults aged 18 years or older residing in Kiambu and Nakuru counties, Kenya, were eligible for inclusion. Trained research assistants with clinical backgrounds recruited participants through household visits. Eligibility criteria included residence in the selected household for at least six months, availability for the study visit, and fluency in either English or Kiswahili, the languages used for all study materials. Only a very small number of potential participants were excluded due to language barriers, as the study counties border Nairobi and the vast majority of residents are conversant in either English or Kiswahili.

### Study design and sampling

The minimum sample size was calculated using Fisher's exact formula, assuming a 50% prevalence of multimorbidity, a 95% confidence level, 2.5% absolute precision, and 10% anticipated nonresponse, yielding a target of 1,691 participants. Sample allocation across counties was proportional to population size based on the 2009 Kenya Population and Housing Census, with 850 participants assigned to Kiambu County and 841 to Nakuru County. The sampling frame was derived from the National Sample Survey and Evaluation Programme maintained by the Kenya National Bureau of Statistics.

A two-stage cluster sampling approach was employed. In the first stage, enumeration areas (EAs) were selected by probability proportional to size sampling from a comprehensive list stratified by county. Each EA consisted of 50–149 households with clearly defined boundaries and demographic data. In the second stage, households within each EA were selected by simple random sampling. Within each household, one eligible adult was randomly chosen using the Kish Grid method to ensure unbiased selection [18].

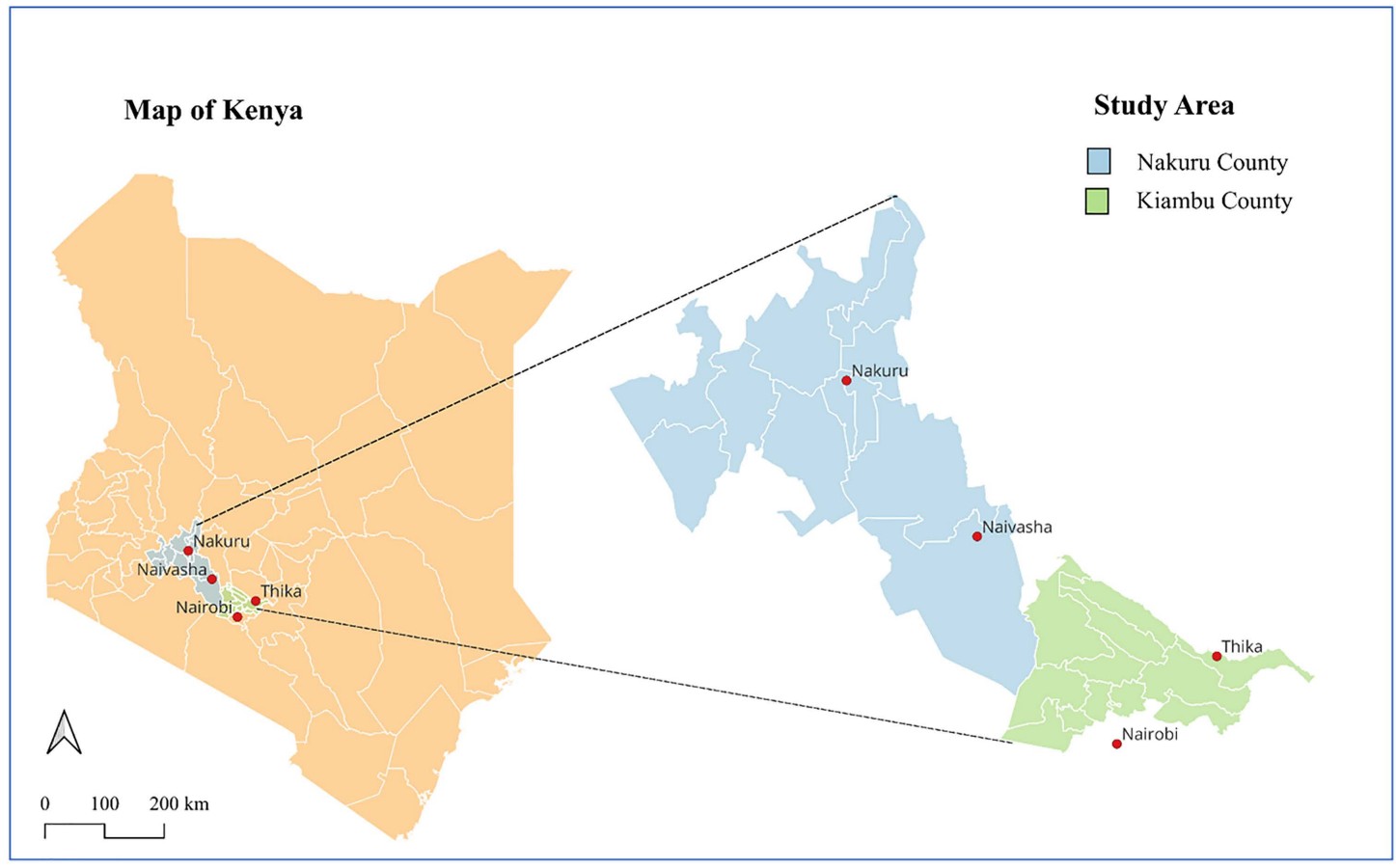

**Fig 1. Map of Kenya showing the various county boundaries and the study areas.** Base layer sources: The Kenya administrative boundary Shape files were retrieved from the Humanitarian Data Exchange website (https://data.humdata.org/dataset/cod-ab-ken) under an open license (https://www.naturalearthdata.com/about/terms-of-use/). Maps were generated using open-source QGIS version 3.34.2 (https://qgis.org/).

### Data collection procedures

The questionnaire was translated into Kiswahili, pretested, and refined before training and fieldwork to ensure clarity and cultural appropriateness [19]. Data were collected between May 2 and July 15, 2017, by trained research assistants. To minimize nonresponse bias, households were revisited if eligible participants were unavailable during the initial visit; no substitutions were made if no one was available after follow-up. For households with eligible members present, one participant was randomly selected using the Kish Grid method, which involved listing all eligible residents by age and sex and applying a pre-assigned random-number table to ensure unbiased selection.

Structured interviews were administered in English or Kiswahili, according to the participant's preference. Questionnaires were pre-programmed in the Open Data Kit (ODK) system on tablets with built-in range and consistency checks to minimize data-entry errors. Written informed consent was obtained prior to the interviews, with participants informed of their right to withdraw at any time or to decline invasive procedures while continuing with the interview.

### Assessment of chronic conditions

Weight was measured to the nearest 0.1 kg using a calibrated digital weighing scale (SECA® 803), and height was measured to the nearest 0.1 cm using a portable stadiometer (SECA® 213), following standardized procedures. Body mass

index (BMI) was calculated as weight in kilograms divided by the square of height in meters. Nutritional status was classified as obese (BMI ≥ 30) or underweight (BMI < 18.5). Blood pressure was measured on the right arm using a validated Omron M6 automated device after participants had rested for at least 5 minutes and abstained from smoking, caffeine, or other stimulants. Three measurements were taken at 5-minute intervals, and the mean value was used for analysis. Random blood glucose was measured from a finger-prick blood sample using a portable point-of-care glucometer (Accu-Chek® Active, Roche Diagnostics), following standardized procedures.

Diabetes was defined as a random blood glucose level ≥11.1 mmol/L or current use of antidiabetic medication. Hypertension was defined as a mean systolic blood pressure ≥140 mm Hg, a mean diastolic blood pressure ≥90 mm Hg, or current use of antihypertensive medication.

Depressive symptoms were assessed with the 9-item Patient Health Questionnaire (PHQ-9), a validated screening instrument for depression [20]. Respondents reported the frequency with which they had experienced the following symptoms two weeks preceding the survey: *'little interest or pleasure in doing things', 'feeling down, depressed or hopeless', 'trouble falling asleep, staying asleep, or sleeping too much', 'feeling tired or having little energy', 'poor appetite or overeating', 'feeling bad about yourself - or that you're a failure or have let yourself or your family down', 'trouble concentrating on things, such as reading the newspaper or watching television', 'moving or speaking so slowly that other people could have noticed or, the opposite - being so fidgety or restless that you have been moving around a lot more than usual, 'thoughts that you would be better off dead or of hurting yourself in some way'*. Responses were scored on a 4-point scale from 0 ("not at all") to 3 ("nearly every day"), yielding a total PHQ-9 score ranging from 0 to 27; scores of 10 or higher, or current use of antidepressant medication, were classified as clinically significant depressive symptoms [20]. All procedures followed standardized protocols to ensure the accuracy, consistency, and reliability of collected data.

## Outcome

The primary binary outcome was multimorbidity, defined as the presence of two or more of the five chronic conditions (obesity, underweight, depression, diabetes, or hypertension).

## Demographic and exposure variables

We collected demographic data on age, sex, education level, employment status, income, and marital status. Age was categorized into four groups: 18–25, 26–35, 36–45, and ≥46 years. Education was classified as primary level or none, secondary level, or tertiary; employment status in the last month as employed or unemployed; and marital status as single, married/cohabiting, separated/divorced, or widowed/widower. Income in the previous month (in Kenyan shillings) was categorized as <1,000; ≥ 1,000–9,999; 10,000–19,999; and ≥20,000.

Substance use was assessed using the WHO Core Model Questionnaire, a standardized instrument designed to capture detailed information on the type and frequency of substance use and related behaviors [19]. Substance use variables included current or prior use of alcohol, tobacco, cannabis, amphetamine-type stimulants, sedatives or hypnotics, tranquilizers, hallucinogens, cocaine, opium, heroin, and volatile inhalants. Use of tobacco products was assessed, including both smoked forms (such as manufactured and hand-rolled cigarettes) and smokeless forms (such as chewing tobacco and snuff). We derived two composite variables: any substance use (use of at least one WHO-defined substance) and poly-drug (multiple) substance use (use of two or more WHO-defined substances)[21]. For each substance, information was collected separately on current use (reporting ongoing use) and on history of use (reporting any prior use, regardless of current use status). This allowed participants to report a history of use even if they were not current users.

## Statistical analysis

We summarized categorical variables as weighted proportions, with standard errors estimated using Taylor series linearization to account for the complex survey design. Analyses were conducted in two stages. First, we examined

socioeconomic inequalities in multimorbidity. Second, we evaluated the association between substance use and multimorbidity after adjustment for all socioeconomic and demographic variables, including age, sex, marital status, education, and income. For categorical variables with more than two levels, we used global Wald tests to assess overall associations.

To estimate adjusted prevalence ratios (aPRs) and 95% confidence intervals (CIs), a weighted modified Poisson regression model was used. Each substance use variable was analyzed separately, adjusting for the aforementioned socioeconomic and demographic factors. Variance inflation factors (VIFs) were calculated to assess multicollinearity among covariates in each model. Substance use categories and combinations were prespecified and were not mutually adjusted, given the expected correlation among them. We conducted sensitivity analyses to examine the association between multimorbidity and a history of substance use, defined as either current use or previous use among individuals who had stopped substance use, potentially for reasons such as medical advice, personal choice, or health concerns. All analyses accounted for the survey design, used a two-sided significance level of 0.05, and were performed with Stata/SE, version 18 (StataCorp) [22]. Because the analyses were exploratory, no corrections for multiple comparisons were applied.

## Results

### Participant characteristics and substance use summaries

A total of 1,484 participants were enrolled, with response rates of 72.9% in Kiambu and 76.9% in Nakuru. The median age of enrolled participants was 36 years (interquartile range IQR: 27–49), 58.3% (n = 859) were female, 60.9% (n = 904) were married, 15.5% (n = 230) had college/university level education, and 15.0% (n = 222) reported earning ≥20,000 Kenya shillings. Overall, 52.6% (n = 780) reported a history of using at least one substance, and 29.1% (n = 432) reported a history of multiple substance use. The most common substances used were alcohol (48.7%) and tobacco (22.6%).

### Prevalence of chronic conditions and multimorbidity

Among the five chronic conditions assessed, 17.7% (n = 250) of participants were obese, 12.6% (n = 183) had hypertension, 7.6% (n = 108) were underweight, 7.0% (n = 104) reported depressive symptoms, and 2.2% (n = 31) had diabetes. Overall, 7.7% (n = 115) met criteria for multimorbidity (≥2 conditions), and only one participant had four conditions. The prevalence of multimorbidity by socioeconomic, demographic, and substance-use characteristics is shown in Tables 1 and 2. The most frequent pairwise combinations of chronic conditions were hypertension and obesity (n = 42), underweight and depression (n = 14), obesity and depression (n = 14), and hypertension and depression (n = 14) (Fig 2).

The prevalence of multimorbidity increased with age, rising from 1.1% (95% CI: 0.2-3.1) among adults aged 18–24 years to 17.8% (95% CI: 14.2-21.9) among those aged 45 years and older (Table 1). No marked sex differences were observed in the overall prevalence of multimorbidity. An analysis of individual chronic diseases revealed a higher prevalence of obesity, hypertension, and diabetes among older age groups (Fig 3). Sex differences were evident only among individuals with obesity, with women experiencing a higher prevalence compared to men (Fig 3).

### Risk factors of multimorbidity among study participants

In multivariable analyses, multimorbidity was significantly more common among older participants (Wald test, 3 df; P < 0.001). Specifically, compared with those aged 18–24 years, the likelihood of multimorbidity was higher among participants aged 36–45 years (aPR 4.32; 95% CI, 1.15–16.28; P = 0.031) and 46 years or older (aPR 12.52; 95% CI, 3.68–42.56; P < 0.001). In addition, the likelihood was higher among females (aPR 1.62; 95% CI, 1.16–2.27; P = 0.005) and participants who were separated or divorced (aPR 2.22; 95% CI, 1.37–3.62; P = 0.002). By contrast, no significant associations were evident among participants who had never married (aPR 0.61; 95% CI, 0.29–1.28; P = 0.191) or those who were widowed (aPR 1.01; 95% CI, 0.52–1.98; P = 0.978). Moreover, there was no evidence of an overall association

**Table 1. Socioeconomic and demographic characteristics of study participants, showing multimorbidity prevalence (with 95% CI) and raw numbers for each factor. Among participants aged 46 years and older, the age ranged from 46 to 99 years, with a median (IQR) of 58 (51–69) years.**

| | Total (N) | Multimorbidity n(%) | (95%CI) |
|---|---|---|---|
| **Overall** | 1484 | 115 (7.7) | (6.4, 9.2) |
| **Age group** | | | |
| 18 - 24 | 278 | 3 (1.1) | (0.2, 3.1) |
| 25 - 35 | 488 | 21 (4.3) | (2.7, 6.5) |
| 36 - 45 | 309 | 18 (5.8) | (3.5, 9.1) |
| 46+ | 410 | 73 (17.8) | (14.2, 21.9) |
| **Sex** | | | |
| Male | 615 | 36 (5.9) | (4.1, 8.0) |
| Female | 859 | 79 (9.2) | (7.3, 11.3) |
| **Marital status** | | | |
| Married | 904 | 67 (7.4) | (5.8, 9.3) |
| Separated/Divorced | 103 | 22 (21.4) | (13.9, 30.5) |
| Never married | 366 | 9 (2.5) | (1.1, 4.6) |
| Widowed | 112 | 18 (16.1) | (9.8, 24.2) |
| **Education level** | | | |
| Primary or less | 719 | 81 (11.3) | (9.0, 13.8) |
| Secondary | 535 | 24 (4.5) | (2.9, 6.6) |
| College/University | 230 | 10 (4.3) | (2.1, 7.9) |
| **Income last month (Kenya Shillings)** | | | |
| <1,000 | 220 | 24 (10.9) | (7.1, 15.8) |
| ≥1,000 to <10,000 | 764 | 59 (7.7) | (5.9, 9.8) |
| ≥10,000 to <20,000 | 278 | 18 (6.5) | (3.9, 10.0) |
| ≥20,000 | 222 | 14 (6.3) | (3.5, 10.4) |

between multimorbidity and either education level (Wald test, 2 df; P = 0.1043) or income in the past month (Wald test, 3 df; P = 0.9657) (Table 3).

In the analysis of substance use, the likelihood of multimorbidity was significantly higher among individuals who reported current alcohol use (aPR 1.68; 95% CI, 1.07–2.64; P = 0.024) and was consistent in a sensitivity analysis that examined a history of alcohol use (aPR 1.65; 95% CI, 1.04–2.62; P = 0.034) regardless of current use. There were no significant associations between multimorbidity and current use of at least one WHO defined substance (aPR 1.14; 95% CI, 0.71–1.83; P = 0.591), cigarette smoking or tobacco use (aPR 0.76; 95% CI, 0.36–1.63; P = 0.481), or current use of multiple substances (aPR 1.09; 95% CI, 0.44–2.66; P = 0.857). These findings were consistent in sensitivity analyses of history of substance use (aPR 1.58; 95% CI, 0.98–2.55; P = 0.059) and history of multiple substance use (aPR 0.76; 95% CI, 0.36–1.62; P = 0.477) (Table 4).

## Discussion

This community-based cross-sectional study found that chronic multimorbidity affected 7% of participants, with prevalence significantly higher among older age groups. In our analysis, older age, being female, and being separated or divorced were significantly associated with a higher burden of multimorbidity, which aligns with much of the international literature. Conversely, no significant associations were found with income or education level. Among the behavioral risk factors examined, only alcohol use showed an association with multimorbidity.

**Table 2. Prevalence of multimorbidity (with 95% CI) by substance use categories among study.**

| | Total (N) | Co-morbidity n (%) | (95%CI) |
|---|---|---|---|
| **Overall** | 1484 | 115 (7.7) | (6.4, 9.2) |
| **Current substance use** | | | |
| Yes | 385 | 28 (7.3) | (4.9, 10.3) |
| No | 1099 | 88 (8.0) | (6.5, 9.8) |
| **History of substance use** | | | |
| Yes | 780 | 67 (8.6) | (6.7, 10.8) |
| No | 704 | 48 (6.8) | (5.1, 8.9) |
| **Current multiple-substance user** | | | |
| Yes | 168 | 9 (5.4) | (2.5, 9.9) |
| No | 1316 | 106 (8.1) | (6.6, 9.7) |
| **History of multiple-substance use** | | | |
| Yes | 432 | 28 (6.5) | (4.3, 9.2) |
| No | 1052 | 87 (8.3) | (6.7, 10.1) |
| **Current tobacco use (smoke cigarettes or chew/sniff tobacco)** | | | |
| Yes | 371 | 27 (7.3) | (4.9, 10.4) |
| No | 1113 | 88 (7.9) | (6.4, 9.7) |
| **Current alcohol use** | | | |
| Yes | 270 | 23 (8.5) | (5.5, 12.5) |
| No | 1214 | 92 (7.6) | (6.2, 9.2) |
| **History of alcohol use** | | | |
| Yes | 723 | 65 (9.0) | (7.0, 11.3) |
| No | 761 | 50 (6.6) | (4.9, 8.6) |

The strong association between age and multimorbidity observed in our study is consistent with findings from diverse populations, where the prevalence of chronic multimorbidity is markedly higher in older age groups [5,23]. This pattern may reflect cumulative exposure to risk factors across the life course, coupled with declining physiologic reserve and increased vulnerability to chronic diseases in later life [24]. Given that multimorbidity in older adults has been linked to functional decline, polypharmacy, and higher mortality, our results highlight the need for age-responsive health care strategies that go beyond single-disease models [25].

We also observed sex differences consistent with prior studies, in which women bear a greater burden of multimorbidity than men [5]. The higher prevalence of multimorbidity among women persisted even after adjusting for age in regression models, suggesting that factors beyond life expectancy. Biological explanations, such as hormonal changes across the reproductive life course and gendered differences in exposure to psychosocial stressors, may account for these differences [26,27]. Consistent with previous studies [28,29], being separated or divorced was associated with higher multimorbidity; marital disruption may elevate chronic disease risk through pathways such as psychosocial stress, reduced social support, and economic instability [30,31]. However, never being married was not associated with higher multimorbidity in our multivariable analyses, consistent with a recent nationally representative study in South Africa [5].

In contrast with studies in some settings [16,32], we found no consistent associations between multimorbidity and income or educational attainment. Several factors may explain this finding. First, the high unemployment rates in Kenya across all education levels may weaken the expected link between education, income, and health, as income in the past month may not fully capture long-term socioeconomic position or material deprivation [33]. Second, widespread informal support systems may buffer the direct effects of low income on health outcomes, further attenuating observed

 

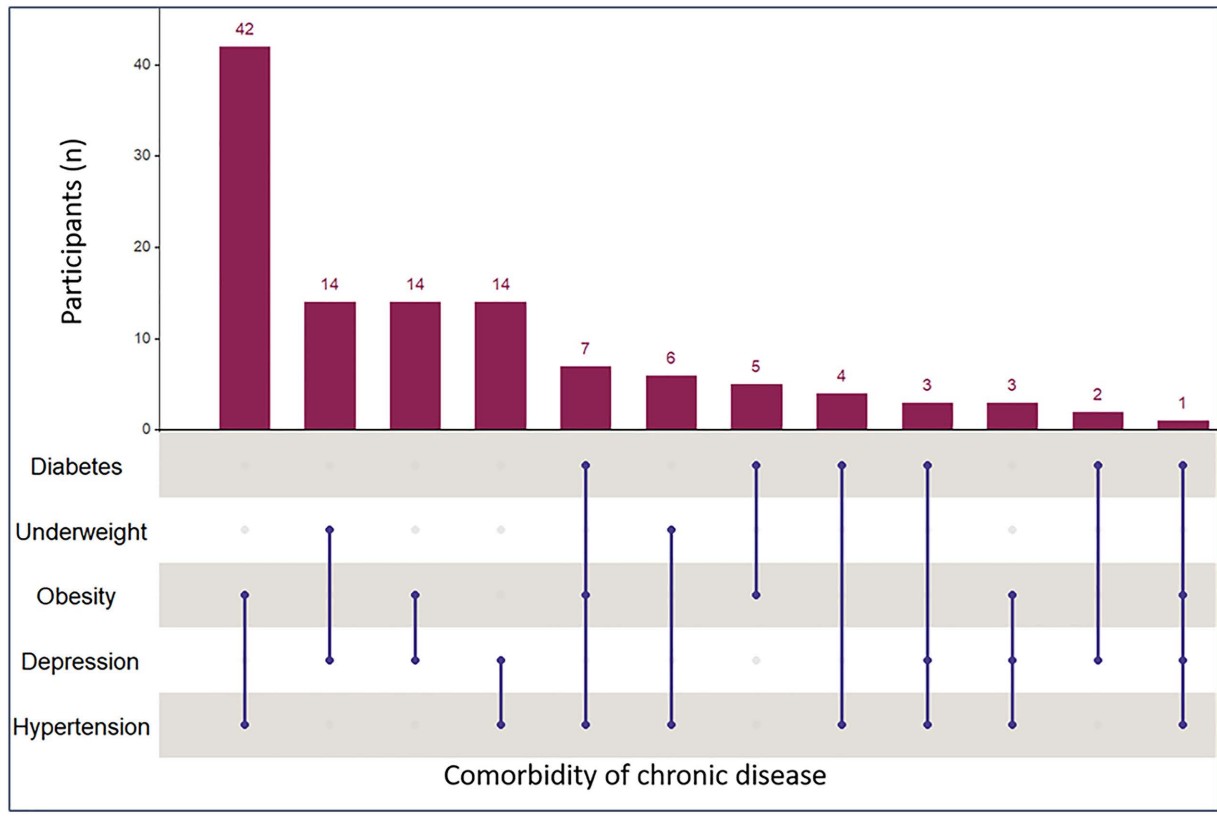

**Fig 2. Frequency of unique combinations of chronic diseases.** Data are unadjusted for the survey design. Filled blue circles along the lines represent the specific chronic disease combinations corresponding to each bar.

associations [34]. Third, studies spanning high-, middle-, and low-income countries have reported a weaker association between multimorbidity and socioeconomic indicators in low-income countries [35]. This suggests that the strength of this relationship is highly context-specific, indicating that the strength and direction of these associations may be context-specific. However, these findings are consistent with a recent study in South Africa showing no association with education or household wealth [5].

Our analysis of behavioral risk factors revealed an association between current alcohol use and multimorbidity, whereas tobacco and other substance use showed no significant associations. Alcohol consumption is a well-established risk factor for hypertension and mental health disorders, among others [36,37]. The significant association between alcohol use and multimorbidity in this population highlights the potential value of integrating alcohol screening and interventions into primary care, particularly in low-resource settings where preventive services are limited.

This study has important limitations. The inclusion of only a small number of chronic conditions likely led to conservative estimates of multimorbidity prevalence and associations, thus limiting the ability to compare prevalences and associations across studies. This limitation reflects a broader challenge in multimorbidity research, which is marked by methodological variability in population selection, sample representativeness, the range of conditions analyzed, and methods of ascertainment [38]. Second, the cross-sectional design precludes causal inference, and third, some measures, particularly income and substance use, relied on self-report and may be subject to recall or social desirability bias. Nonetheless, our study has several important strengths: unlike most prior multimorbidity research, which is typically hospital-based and not representative of the broader population, this analysis was population-based, included both urban and rural

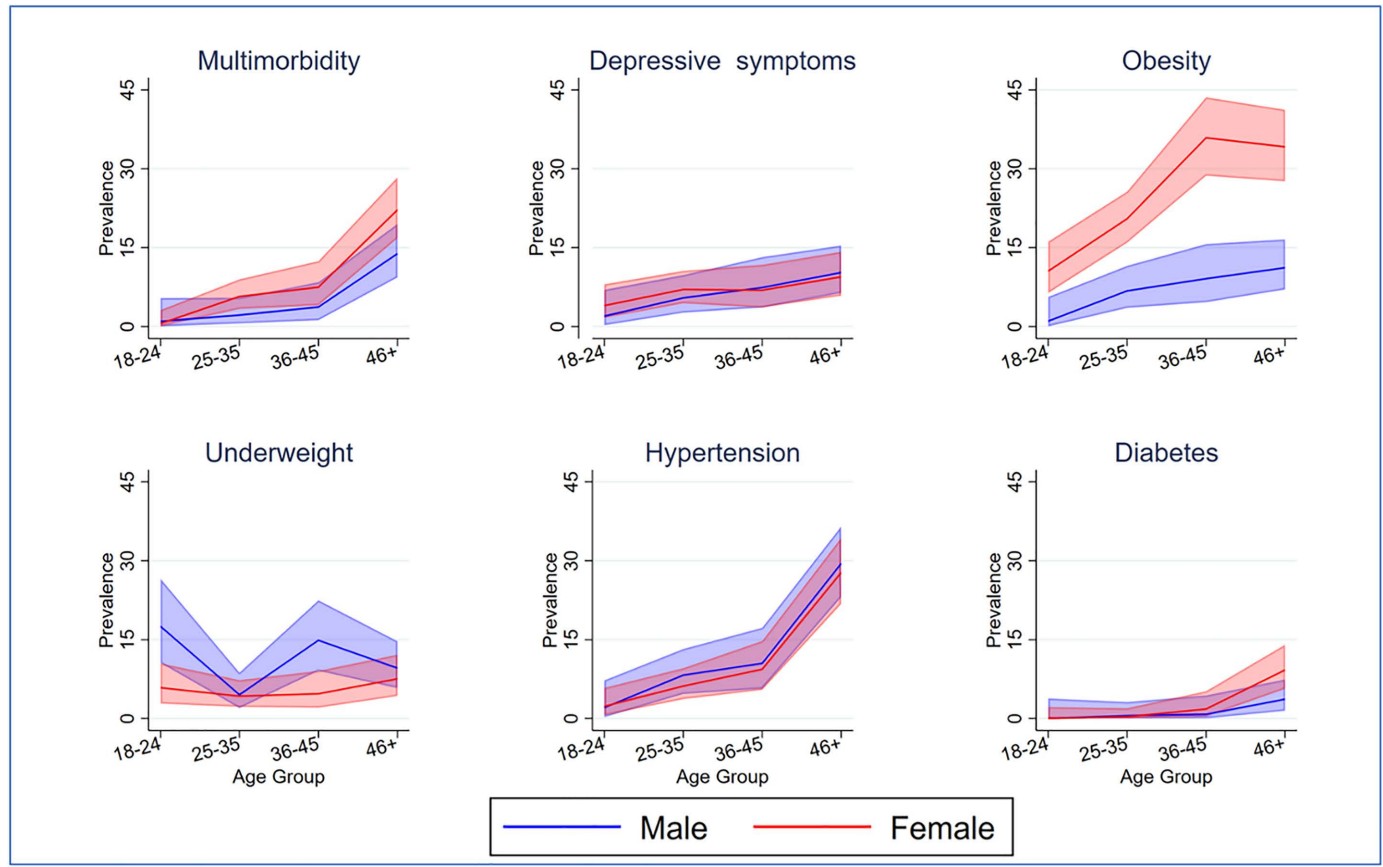

**Fig 3. Prevalence of multimorbidity and chronic vconditions by age (adults ≥18 years) in two Kenyan counties, with 95% confidence intervals.**

settings, and used clinically verified diagnoses rather than solely self-reported ones, thereby enhancing the validity and generalizability of the findings.

## Recommendations

Our findings have important implications for health systems in low- and middle-income countries facing rising burdens of chronic multimorbidity [1]. Age-responsive models of care are needed to address the clustering of conditions in older adults. Gender-sensitive interventions should prioritize women's health across the life course, while providing psychosocial support for individuals experiencing marital disruption. Community sensitization programs aimed at increasing awareness of chronic disease prevention and early detection could further strengthen these efforts. Finally, integrating alcohol-use screening and interventions into primary care may offer a feasible strategy for reducing the burden of multimorbidity.

## Supporting information

**S1 Data. Dataset for replication of analysis.** This file contains anonymized dataset used in the study, including variables necessary to reproduce the statistical analyses presented in the manuscript.
(XLSX)

**Table 3. Association between socio-economic and demographic factors, and multimorbidity among study participants. Model 1 presents results from the univariable modified Poisson regression model, and Model 2 presents results from a multivariable modified Poisson regression model. Among participants aged 46 years and older, the age ranged from 46 to 99 years with a median (IQR) of 58 (51–69) years.**

| | Model 1: Univariable Analysis | | | Model 2: Multivariable Analysis | | |
|---|---|---|---|---|---|---|
| | PR | 95% CI | p-value | aPR | 95% CI | p-value |
| Age group (ref. 18–24) | | | | | | |
| 25 - 35 | 4.77 | (1.29,17.63) | 0.020 | 3.35 | (0.88,12.82) | 0.076 |
| 36 - 45 | 6.29 | (1.77,22.43) | 0.005 | 4.32 | (1.15,16.28) | 0.031 |
| 46+ | 19.38 | (6.07,61.87) | 0.000 | 12.52 | (3.68,42.56) | 0.000 |
| Sex (ref. Male) | | | | | | |
| Female | 1.57 | (1.11, 2.24) | 0.012 | 1.62 | (1.16, 2.27) | 0.005 |
| Marital Status (ref. Married) | | | | | | |
| Separated/Divorced | 2.89 | (1.68, 4.98) | 0.000 | 2.22 | (1.37, 3.62) | 0.002 |
| Never married | 0.32 | (0.16, 0.63) | 0.001 | 0.61 | (0.29, 1.28) | 0.191 |
| Widowed | 2.19 | (1.13, 4.22) | 0.020 | 1.01 | (0.52, 1.98) | 0.978 |
| Education (ref. primary or less) | | | | | | |
| Secondary | 0.39 | (0.23, 0.67) | 0.001 | 0.58 | (0.35, 0.96) | 0.035 |
| College/University | 0.40 | (0.20, 0.81) | 0.011 | 0.75 | (0.36, 1.58) | 0.451 |
| Income last month (Kenya Shillings)(ref. <1000) | | | | | | |
| >=1,000 to <10,000 | 0.70 | (0.40, 1.24) | 0.215 | 1.08 | (0.61, 1.94) | 0.782 |
| >=10,000 to <20,000 | 0.59 | (0.31, 1.12) | 0.105 | 1.16 | (0.60, 2.24) | 0.658 |
| >20,000 | 0.57 | (0.27, 1.19) | 0.131 | 1.18 | (0.52, 2.71) | 0.689 |

*ref =reference group

**Table 4. Association between substance use and multimorbidity among the study participants. Model 1 presents results from the univariable modified Poisson regression model, and Model 2 presents results from a multivariable modified Poisson regression model. In the multivariable analyses, each substance use variable has been adjusted for age, sex, marital status, education, and income.**

| | Model 1: Univariable Analysis | | | Model 2: Multivariable Analysis | | |
|---|---|---|---|---|---|---|
| | PR | 95% CI | p-value | aPR | 95% CI | p-value |
| Current substance use (ref. No) | | | | | | |
| Yes | 0.90 | (0.57, 1.42) | 0.655 | 1.14 | (0.71, 1.83) | 0.591 |
| History of substance use (ref. No) | | | | | | |
| Yes | 1.27 | (0.84, 1.94) | 0.255 | 1.58 | (0.98, 2.55) | 0.059 |
| Current multiple substance use (ref. No) | | | | | | |
| Yes | 0.69 | (0.30, 1.55) | 0.360 | 1.09 | (0.44, 2.66) | 0.857 |
| History of multiple substance use (ref. No) | | | | | | |
| Yes | 0.78 | (0.49, 1.24) | 0.297 | 0.76 | (0.36, 1.62) | 0.477 |
| Current tobacco use [smoke cigarettes or chew/sniff tobacco] (ref. No) | | | | | | |
| Yes | 0.80 | (0.41, 1.54) | 0.491 | 0.76 | (0.36, 1.63) | 0.481 |
| Current alcohol use (ref. No) | | | | | | |
| Yes | 1.11 | (0.68, 1.82) | 0.672 | 1.68 | (1.07, 2.64) | 0.024 |
| History of alcohol use (ref. No) | | | | | | |
| Yes | 1.37 | (0.92, 2.04) | 0.119 | 1.65 | (1.04, 2.62) | 0.034 |

*ref= reference group

## Acknowledgments

We thank David Mathu and Rodgers Ochieng for coordinating field activities; the research assistants for their dedication and support; and the Kenya National Bureau of Statistics for assistance with sampling, household selection, and data weighting. We are especially grateful to all study participants for their time and contributions.

## Author contributions

**Conceptualization:** Linnet Ongeri, Lydia Kaduka, Damaris Matoke, Doris Njomo, Zipporah Bukania, Moses Mwangi, Linus Ndegwa, Joanna Olale, Caleb Othieno.

**Data curation:** Linnet Ongeri, Lydia Kaduka, Damaris Matoke, Zipporah Bukania, Moses Mwangi, Geoffrey Barini, Valentine Wanga, Polycarp Mogeni.

**Formal analysis:** Moses Mwangi, Geoffrey Barini, Valentine Wanga, Polycarp Mogeni.

**Funding acquisition:** Linnet Ongeri.

**Methodology:** Linnet Ongeri, Lydia Kaduka, Damaris Matoke, Doris Njomo, Zipporah Bukania, Moses Mwangi, Linus Ndegwa, Joanna Olale, Caleb Othieno, Sahara Hussein, Geoffrey Barini, Valentine Wanga, Polycarp Mogeni.

**Project administration:** Sahara Hussein.

**Supervision:** Linnet Ongeri, Polycarp Mogeni.

**Validation:** Lydia Kaduka, Damaris Matoke, Doris Njomo, Linus Ndegwa.

**Writing – original draft:** Linnet Ongeri.

**Writing – review & editing:** Linnet Ongeri, Lydia Kaduka, Damaris Matoke, Doris Njomo, Zipporah Bukania, Moses Mwangi, Linus Ndegwa, Joanna Olale, Caleb Othieno, Sahara Hussein, Geoffrey Barini, Valentine Wanga, Polycarp Mogeni.

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
