## [Decision Letter · Decision Letter 0]

18 Aug 2025

PGPH-D-25-01204

Multimorbidity of Chronic Diseases and Association with Substance Use in Kiambu and Nakuru Counties in Kenya

Dear Dr. Ongeri,

Thank you for submitting your manuscript to PLOS Global Public Health. After careful consideration, we feel that it has merit but does not fully meet PLOS Global Public Health’s publication criteria as it currently stands. Therefore, we invite you to submit a revised version of the manuscript that addresses the points raised during the review process.

We look forward to receiving your revised manuscript.

Kind regards,

Prof Razak Gyasi, PhD, PD

Academic Editor

Journal Requirements:

1. We ask that a manuscript source file is provided at Revision. Please upload your manuscript file as a .doc, .docx, .rtf or .tex.

2. Please send a completed 'Competing Interests' statement, including any COIs declared by your co-authors. If you have no competing interests to declare, please state "The authors have declared that no competing interests exist". Otherwise please declare all competing interests beginning with the statement "I have read the journal's policy and the authors of this manuscript have the following competing interests:"

3. In the online submission form, you indicated that “Data used in this analysis will be published online once all protocol objectives are completed. Reasonable requests for data can be directed to the corresponding author.”.

a. In a public repository,

b. Within the manuscript itself, or

c. Uploaded as supplementary information.

4. Some material included in your submission may be copyrighted. According to PLOS’s copyright policy, authors who use figures or other material (e.g., graphics, clipart, maps) from another author or copyright holder must demonstrate or obtain permission to publish this material under the Creative Commons Attribution 4.0 International (CC BY 4.0) License used by PLOS journals. Please closely review the details of PLOS’s copyright requirements here: PLOS Licenses and Copyright. If you need to request permissions from a copyright holder, you may use PLOS's Copyright Content Permission form.

Potential Copyright Issues: Figure 1: please (a) provide a direct link to the base layer of the map (i.e., the country or region border shape) and ensure this is also included in the figure legend; and (b) provide a link to the terms of use / license information for the base layer image or shapefile. We cannot publish proprietary or copyrighted maps (e.g. Google Maps, Mapquest) and the terms of use for your map base layer must be compatible with our CC-BY 4.0 license.

Additional Editor Comments (if provided):

Reviewers' comments:

Reviewer's Responses to Questions

**Comments to the Author**

1. Does this manuscript meet PLOS Global Public Health’s publication criteria?

Reviewer #1: Yes

Reviewer #2: Partly

Reviewer #3: Yes

2. Has the statistical analysis been performed appropriately and rigorously?

Reviewer #1: N/A

Reviewer #2: No

Reviewer #3: Yes

3. Have the authors made all data underlying the findings in their manuscript fully available (please refer to the Data Availability Statement at the start of the manuscript PDF file)?

Reviewer #1: Yes

Reviewer #2: No

Reviewer #3: Yes

4. Is the manuscript presented in an intelligible fashion and written in standard English?

Reviewer #1: No

Reviewer #2: No

Reviewer #3: Yes

Reviewer #1: R1:

fine and clear topic but needs great effort to be obvious.

don't use pronouns in research language.

use mean full and brief key words 3-5words as maximum

abstract is presented in clear waY, but some information were missed.

Please give short background in abstract.

Introduction:

Author can numerate the multi-morbidity disease in Kenya.

Focus on multi-morbidity disease of the current study.

The end of introduction should contains complete idea of study problem

The study problem should be similar to that of aim of the study in abstract

avoid to use pronouns like we.

Methodology:

methodology needs great efforts to be acceptable

Study methodology contains some literature that defect the content of the methods

Methods depends on restricted way that describe the practical and the way of conducting research.

please summaries and re-write study methods in brief manner include the following:

1. type of the study

2. description of the participants

3. methods of collecting data

4. method of sample collection

5. inclusion and exclusion criteria of study participants

6. divide participants into groups according to the study objects.

7. data analysis.

8. no need for many details that hide the main information about the exact title.

9. study outcomes is related to the expected results.

Results:

prefer to use a table beside text.

put every table next to its text.

Discussion:

fine. But long discussion. Focus on main and significant results.

divide the discussion of the results from limitation, recommendations and conclusion.

Conclusions is restricted towards the study object.

Reviewer #2: The authors have completed a population-based survey in two counties in Kenya and used the resulting data to explore the cross-sectional associations alcohol and other substances of abuse and multimorbidity. I congratulate the authors on the completion of the survey, a rare piece of important data from Kenya and across the region. The current analysis has a number of substantial limitations and interpretations require some rethinking.

• The abstract lacks a definition for “substance use” and “multimorbidity” given that the precise definition is critical to understand the results, these need to be clearly defined here.

• The abstract does not specify whether the estimates shared come from univariate or multivariate models

• The abstract uses the term “female gender” but gender was not measured by the authors. The term is female sex.

• The proportion of married individuals was misstated as 60% when in the text it was reported to be 61%

• On line 52 the authors write “Multimorbid conditions focus on a holistic…” Conditions don’t “focus” rather the users of the multimorbidity concept may “focus.”

• The authors claim based on reference 2 that multimorbidity is growing, which presumably meets the rates are increasing. Reference 2 does not make that claim.

• The use of the word “carry” on liner 57 is incorrect.

• There are 2 full pages of introduction and much of the material is not directed at the subject at hand. The background should be shortened and focus only on the prevalence and predictors of the examined multimorbid conditions, and it should be restricted mainly to Kenya. For example, there is no need to give special consideration to people living with HIV as that is not what this papers aims to explore.

• Line 85 refers to the prevalence of depression in patient populations, but it isn’t clear what kind of patients they are discussing.

• On line 135 they say that the survey was limited to subjects that spoke English or Kiswahili. They should share the fraction of subjects that were excluded on this basis.

• On line 157 they say that they took actions to minimize non-response bias, but they don’t report the response rate. It should be here.

• On lines 172-177 they describe data reclassification, but they have yet to explain where that data would come from, so the methods are mis-ordered. Furthermore, they describe classifying fasting blood sugars, but I don’t know where that data would come from as on line 182 they say they assesses “Random Blood Sugar” levels but there is no indication how that testing was completed.

• The definition of multimorbidity given on lines 198-199 needs to be in the abstract

• On line 220 in the methods, the authors claim they use a backwards stepwise model, but I see no evidence of this in the results (e.g. there is nothing about that in Table 4).

• On lines 242-244 they detail the prevalence of the underlying conditions that are summed to make multimorbidity only after they have reported the summed variable prevalence. That is out of order.

• In Tabel 1, the top age range is described as 46+, that is not enough information. The actual range and the mean age in that range need to be shared somewhere in the paper.

• Table 2 uses many different ways of saying “using” a substance. All of these should be converted to use, not “on” or “taking”.

• The authors don’t explain what forms of tobacco are used other than cigarettes and the phrase “cigarettes or tobacco” is confusing since cigarettes are tobacco products.

• Testing and presenting correlations between a summed variable (multimorbidity) and the underlying conditions does not make sense. They should describe the prevalence of each, but a correlation is not justified.

• The statement on Line 277 makes no sense, by definition current users are ever users. I wonder what they were trying to convey and if they understand their data?

• On line 278, they use the word interesting which is conversational, an editorial comment, and does not belong in the results section of the paper.

• Table 3 is in fact a figure.

• The authors have not completed the proper test of association for education. To test for an education effect, they need to conduct a 2 df test on the association of the two categories with the outcome and only that can determine if there is an overall education effect.

• The line numbers disappeared on page 14.

• The authors make the claim on page 14/15 that there is an association among those using 3 or more substances, but the P is 0.89. That means there is nothing there.

• In the second paragraph of the discussion, the authors make odd statements about what biological aging is. This has nothing to do with their work and does not belong in the paper. And there is no such thing as “slowing the aging process.”

• The authors again use the word gender on page 17 but haven’t measured gender.

• The authors make an incorrect claim about the effect of being widowed. There is no association with being widowed in the full model.

• The authors use reference 64 to make very speculative claims about the educational effects that are not warranted.

• The authors claim that their data shows that alcohol has a “unique and substantial impact on chronic disease development” on page 17. That is not correct. Their data can be used to apportion cross-sectional associations in their population but not make grand claims about all the other substances of abuse.

• The authors seem to misinterpret the role of multivariate models at the top of page 18 as they describe confounding effects of tobacco etc on alcohol consumers. The point of the models was to separate those effects, so why revert to describing it this way?

• They go on to speculate about the need for “brief interventions” and treatment despite discussing ever use of alcohol! Ever users of alcohol do not need interventions or treatment. They haven’t assessed Alcohol Use Disorder, which might suggest what needs to be done to improve population health.

• The conclusion makes a claim about “comparable studies” but I don’t know what that is referring to. To make comparisons they need to have age-standardized rates and ideally, they would compare their data to other LMIC populations.

• They also make an incorrect claim about “higher levels of education” which is not backed by the results they presented.

• The data availability statement seems inadequate since it makes a vague promise to post the data at some future date. A more precise plan should be shared.

Reviewer #3: Thank you for the opportunity to review this paper.

The authors address a key social health problem and are commended for their work.

I have a few suggestions.

Introduction

1. Mention the measures Kenya is taking with the recent allocation of the emergency and chronic disease fund scheme under the Social Health Authority and cite accordingly in the introduction

2. While the authors mention paucity of data on the topic, there is actually some work done in Kenya on the general topic of substance use. Consider adding local Kenyan studies to set a good introduction and local data.

• https://journals.plos.org/plosone/article?id=10.1371/journal.pone.0269340

• https://pubmed.ncbi.nlm.nih.gov/35300637/

• https://journals.plos.org/globalpublichealth/article?id=10.1371/journal.pgph.0004130

3. Similarly, cancer wasn’t mentioned in the introduction as a leading killer disease in Kenya with regard to chronic disease; therefore, it should be noted at least as a limitation. So is HIV status among such a youthful cohort of study participants.

**Do you want your identity to be public for this peer review?** For information about this choice, including consent withdrawal, please see our Privacy Policy

Reviewer #1: **Yes: ** NAHLA AHMED MOHAMMED ABDERHMAN

Reviewer #2: No

Reviewer #3: **Yes: ** Omar Abdihamid MD

---

## [Decision Letter · Decision Letter 1]

12 Nov 2025

Socioeconomic inequalities, substance use, and chronic multimorbidity in Kiambu and Nakuru counties in Kenya

PGPH-D-25-01204R1

Dear Dr. Ongeri,

We are pleased to inform you that your manuscript 'Socioeconomic inequalities, substance use, and chronic multimorbidity in Kiambu and Nakuru counties in Kenya' has been provisionally accepted for publication in PLOS Global Public Health.

Best regards,

Julia Robinson

Executive Editor

Reviewer Comments (if any, and for reference):

Reviewer's Responses to Questions

**Comments to the Author**

Reviewer #2: All comments have been addressed

publication criteria?

Reviewer #2: Yes

3. Has the statistical analysis been performed appropriately and rigorously?

Reviewer #2: Yes

4. Have the authors made all data underlying the findings in their manuscript fully available (please refer to the Data Availability Statement at the start of the manuscript PDF file)?

Reviewer #2: (No Response)

5. Is the manuscript presented in an intelligible fashion and written in standard English?

Reviewer #2: Yes

Reviewer #2: The authors have addressed all my concerns and made appropriate updates to the manuscript.

**Do you want your identity to be public for this peer review?** For information about this choice, including consent withdrawal, please see our Privacy Policy

Reviewer #2: No
